# Simulation of the Transient Characteristics of Water Pipeline Leakage with Different Bending Angles

**Qiaoling Zhang \*, Feng Wu, Zhendong Yang, Guodong Li and Juanli Zuo**

State Key Laboratory of Eco-hydraulics in Northwest Arid Region of China, Institute of Water Resources and Hydro-Electric Engineering, Xi'an University of Technology, Xi'an 710048, China
\* Correspondence: zqling0712@163.com; Tel.: +86-029-8231-2788

**Abstract:** Rapid global development has resulted in the widespread use of water pipelines in industrial and agricultural production and life. During water transportation and deployment, water pipes with different angles need to be positioned according to different geographical and topographical problems. Flowmaster simulation software was used to simulate the leakage process of pipelines with different angles. The transient characteristics of fluids in the pipeline were studied in detail. The influences of parameters, such as the bending coefficient R/D (R is the turning radius of pipe, D is the inner diameter of pipe), leakage position, and leakage aperture on the transient flow law of pipelines with and without leakage, were analyzed. The results show that the periodic decay of the upstream flow and pressure curve at the valve with and without leakage has an insignificant relationship with the bending coefficient R/D; however, the amplitude of the sudden position change is positively correlated with the magnitude of R/D. The leakage aperture is positively correlated with the leakage flow and negatively correlated with the pressure value at the leak location node. The farther the leak position is from the valve, the greater the amplitude of the valve end pressure and the upstream flow curve, and symmetric fluctuations occur.

**Keywords:** bending coefficient; leakage aperture; leakage position; Flowmaster simulation; leakage detection

---

## 1. Introduction

With the rapid development of production and living standards in recent years, water pipelines have become the backbone of water resource deployment and transportation. An actual project needs to decorate different angles of a bending pipe to adapt to a variety of geographical and topographical environments. However, pipeline leakage is inevitable in the long-distance water transmission process. If a leak cannot be identified and processed promptly, it can cause severe economic losses or environmental pollution. Therefore, the safety of water pipelines has become the focus of several studies.

Hitherto, many scholars have conducted research on water pipeline leakage detection and location. Pipeline leakage detection technology has been the focus in recent years; however, the transient flow detection method [1,2] still occupies a pivotal position. To determine the existence of pipeline leakage, Liu et al. [3] described an integrated model for leakage detection and location, which can be used to identify micro-leakages in liquid pipelines, including almost all leakages. Brunone et al. [4] used direct transient analysis (DTA) to directly identify defects in the pressure signal that were allowed to pass based on the analysis of the damping of the pressure peak. It was found that the pressure decay index in the polymer leakage tube depends on the size and location of the leak and the leak pressure. Wang et al. [5,6] discriminated the leakage position and leakage quantity using the fast Fourier transform to further determine the time domain problems related to pipeline leakage

under variable conditions and found that the speed of the water strike wave amplitude decreases during sudden shutdown of the valve. Li et al. [7] proposed a novel location algorithm based on the attenuation of negative pressure wave (ANPW) to accurately determine the location of the leakage. Ferrante et al. [8,9] used a pulse signal generated by instantaneously closing the valve to determine the pressure at the end of a pipe in the frequency domain, and the wavelet transform method was used for signal processing. Witness et al. [10] formed a pressure frequency response diagram of the valve end. It was found that the amount of leakage and the leakage position are related to the primary and secondary pressure resonance peaks. The secondary pressure resonance peak appeared in the frequency plot with leakage and was larger than the amplitude of the main pressure amplitude without leakage. Gong et al. [11,12] determined the location and size of the leak based on the frequency response and analyzed the effects on leakage detection of the odd and even signals of the frequency response map. Kim [13] deduced the frequency domain response function based on the impedance method and analyzed the influences of parameters such as leakage, friction coefficient, wave velocity, pipe length, and valve closing time. He and Ayadi [14,15] constructed a mathematical model of the pipeline transient flow, and the influence of the unbalanced friction was considered. Particularly in real pipe systems, the technique used to generate pressure waves plays a crucial role. Shucksmith et al. [16] proposed a pressure transient leak location technique based on analyzing pressure waves reflected by leaks and features in pipes and accurately identifying small pressure waves caused by pipe feature (including leaks) reflection. This technique requires accurate estimation of the wave propagation speed. This method has the potential to increase the speed and accuracy of leak locations and reduce the occurrence of inaccurate leak diagnostics. Lee et al. [17] verified the experiment on the frequency response method of pipeline leakage detection and found that when the pipeline leaks, a periodic pattern is generated on the resonance peak of the frequency response diagram. This mode can be used as an indicator of leakage without the need to compare it to a "no leak" baseline.

　　With the rise of computer technology and other related disciplines, the pipeline leakage detection problem has evolved in the direction of combining multiple detection methods in order to better utilize the strengths and advantages of various methods. Ge et al. [18] proposed a method to evaluate pipeline sensitivity and minimum leakage and analyzed the effects of pressure wave attenuation and amplitude variation due to pipeline leakage. The research results show that the accuracy of the instrument and external environment, including the characteristics of the pipeline itself, the pressure of the inlet and outlet, and the detection of the minimum leakage of the fluid, has a great impact. Liang and Ning [19] proposed a leak location method based on integrated pressure and flow signals that could detect slow leaks and small leaks. Meseguer et al. [20] proposed a model based on a methodology for line leakage detection and described the localization. Lu et al. [21] proposed a small noise reduction method based on empirical mode decomposition (SNR-EMD) which reduces noise in the pipeline pressure signal. The case study indicated that pressure drop can be well-identified, and leakage can be accurately located. Diao et al. [22] proposed a modified transient-based method for leakage detection and location in a reservoir-pipe-valve (RPV) system. The leak location model was based on the time when a pressure wave propagated from the valve to the leakage location and back again, and it showed high accuracy and great performance in leakage detection and location. Li et al. [23] experimentally researched the leakage detection of a water distribution system that was subjected to socket joint failure using acoustic emission (AE) techniques. Furthermore, the acoustic characteristics of leak signals in the socket and spigot pipe segments were investigated. Cheng et al. [24] used an effective reasoning method to deal with the uncertain information to determine whether a leakage had occurred or not. He used a hydraulic model to determine the location of the leakage, and the proposed leakage detection system had high reliability. Cheng et al. [25] considered the accuracy of leakage detection and other important factors and recognized the close relationship between pipeline operation noise and detection accuracy. Shao et al. [26] considered the limitations of the classical orifice plate equation and studied the effect of the main flow velocity on leakage. Al-Washali et al. [27] studied a method to reduce the leakage through the analysis of the minimum night flow. Adedeji et al. [28] proposed a suitable algorithm

for the water distribution network model, which had a strong applicability to pipe networks. In fact, the presence of many branches in the water distribution network (WDN) makes it very difficult to use transient test-based techniques (TTBTs) [29,30]. Meniconi et al. [30] proposed a large difference between the transmission main power (TM) and the WDN, and the water mains had certain limitations. In TM detection, the most common fault detection technology is the inline type, as the insertion of pipe sensors is more expensive than the sensors used in distribution networks (DNs).

To achieve a deeper understanding of the instantaneous process characteristics of water pipeline leakage, this study uses Flowmaster simulation software to calculate the leakage of a water pipeline and analyze the influence of the bending coefficient R/D (R is the turning radius of pipe, D is the inner diameter of pipe), different leakage holes, and leakage position on the transient flow law of the pipeline when there is no leakage. The variations in the hydraulic characteristics of a straight pipe, 90° and 180° bent pipes under the condition of leakage are compared and analyzed, providing theoretical guidance for the practical application of transient flow leakage detection.

## 2. Numerical Model

### 2.1. Governing Equations

In this study, the corresponding transient flow equations were established to solve the transient flow signal with the help of the Flowmaster software under certain boundary conditions. The flow field parameters such as the pressure and flow rate of the pipeline were obtained, and the leakage amount and leakage location were determined in order to detect and locate the pipeline leakage.

The basic mathematical equations of the water hammer transient flow in the pipeline are as follows:

$$\frac{\partial H}{\partial t} + v\frac{\partial H}{\partial x} + \frac{a^2}{g}\frac{\partial v}{\partial x} = 0$$
$$\frac{\partial H}{\partial x} + \frac{1}{g}\frac{\partial v}{\partial t} + \frac{v}{g}\frac{\partial v}{\partial x} + J_S + J_U = 0 \tag{1}$$

where $L_1$ is the continuity equation, $H$ is the fluid head, $t$ is the time, $a$ is the water hammer wave propagation velocity, $g$ is the gravity acceleration, $v$ is the velocity, and $x$ is the distance along the pipe. $L_2$ is the equation of motion, $J_S$ is the steady friction, and $J_U$ is the unsteady friction.

### 2.2. Non-Steady Friction Model

The Brunone model, which combines the instantaneous time-varying acceleration and potential variable acceleration, can better adapt to the transient flow model and has higher accuracy in both laminar flow and turbulence:

$$J_U = \frac{k}{g}\left(\frac{\partial v}{\partial t} - a\frac{\partial v}{\partial x}\right) \tag{2}$$

$$k = \frac{\sqrt{C^*}}{2}, C^* = \frac{7.41}{\text{Re}^{k'}}, k' = \log_{10}\left(\frac{14.3}{\text{Re}^{0.05}}\right) \tag{3}$$

where $k$ is the Brunone coefficient, Re is the Reynolds number, and $J_U$ is non-steady friction.

The steady friction can be expressed as

$$J_s = \frac{fV|V|}{2gD} \tag{4}$$

where $D$ is the pipe diameter, Re is the Reynolds number, $f$ is the Darcy–Weisbach friction coefficient, and $J_S$ is the steady friction coefficient.

### 2.3. Method of Characteristics

We ignore the convective terms of $v$ and $H$ in the equation, because $a >> |v|$. The flow rate $v$ is replaced with the pipe flow rate $Q$ in Formula (5). The method of characteristics (MOC) is used

to transform it into ordinary differential equations, and the positive and negative characteristic line equations are obtained as follows:

$$C^+ : dH + \frac{a}{gA}dQ + \frac{fa}{(1+m)D}\frac{Q|Q|}{2gA^2}dt = 0 \quad \frac{dx}{dt} = \frac{a}{1+m}$$
$$C^- : dH - \frac{a(1+m)}{gA}dQ - \frac{fa}{D}\frac{Q|Q|}{2gA^2}dt = 0 \quad \frac{dx}{dt} = -a \tag{5}$$

Let $B = \frac{a}{gA}$, $R = \frac{fa\Delta t}{2gA^2D}$, $Q|Q| = Q_P|Q_R|$, $C_8 = 1 + m$, be integrated along the $C^+$ and $C^-$ characteristic lines of the difference grid shown in Figure 1, respectively, and the finite difference equation is obtained as follows:

$$C^+ : H_P - H_R + B(Q_P - Q_R) + R/C_8 Q_P|Q_R| = 0$$
$$C^- : H_P - H_B - BC_8(Q_P - Q_B) - RQ_P|Q_B| = 0 \tag{6}$$

The pressure head $R$ and flow parameter $Q_R$ of the point in the above formula can be obtained by $A$ and $C$ two-point linear interpolation, which can be simplified as follows:

$$\begin{aligned} C^+ : H_P &= (H_R + BQ_R) - (B + R/C_8 \times |Q_R|) \times Q_P \\ &= A_1 - A_2 Q_P \\ C^- : H_P &= (H_B - B \times C_8 Q_B) + (B \times C_8 + R|Q_B|)Q_P \\ &= D_1 + D_2 Q_P \\ A_1 &= H_R + BQ_R, \ A_2 = B + R/C_0|Q_R| \\ D_1 &= H_B - B \times C_0 Q_B, \ D_2 = BC_0 + R|Q_B| \end{aligned} \tag{7}$$

By combining the difference equations in the $C^+$ and $C^-$ directions, the hydraulic parameters of nodes without leakage can be obtained:

$$Q_P = \frac{A_1 - D_1}{A_2 + D_2} H_P = \frac{A_1 + D_1 + (D_2 - A_2)Q_P}{2} \tag{8}$$

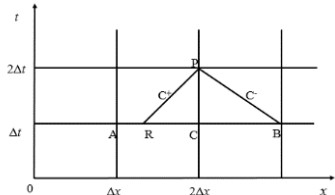

**Figure 1.** Characteristic line difference grid.

## 2.4. Boundary Condition

### 2.4.1. Leakage Hole Boundary Condition

Let $Q_L^+$ and $Q_L^-$ be the positive and negative flow rates of the leak hole. As shown in Figure 2. $C_g$ is the leakage hole coefficient. Assuming that outside the leakage hole is atmospheric pressure, the characteristic line equation before and after the leakage point can be obtained according to basic mathematical equations:

$$C^+ : H_L = A_1 - A_2 Q_L^+$$
$$C^- : H_L = D_1 + D_2 Q_L^- \tag{9}$$

If the leakage point is regarded as a small hole outflow, the leakage flow rate $Q_L$ is

$$Q_L = C_g \sqrt{2gH_L} \tag{10}$$

From the continuity theorem of traffic, we can see that

$$Q_L^+ = Q_L^- + Q_L \tag{11}$$

The pressure $H_L$ and flow rate $Q_L$ of the leakage point are obtained as follows:

$$H_L = \frac{D_2 A_1 + A_2 D_1 - A_2 D_2 D_L}{D_2 + A_2}$$
$$Q_L^+ = \frac{A_1 - H_L}{A_2}, Q_L^- = \frac{H_L - D_1}{D_2} \tag{12}$$

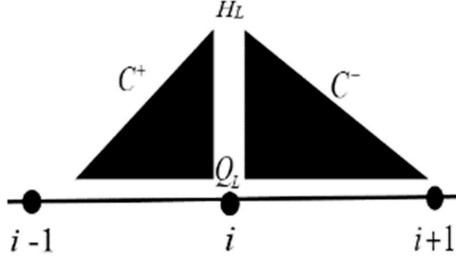

**Figure 2.** Leakage hole boundary condition.

### 2.4.2. Upstream Boundary Condition

It is known that the upper reaches are a reservoir, and the water level $H_0$ of the reservoir remains fixed. $H_0$ is brought into the $C^-$ characteristic line equation, and the solution is obtained.

$$Q_{P1} = \frac{H_0 - D_1}{D_2} \tag{13}$$

### 2.4.3. Downstream Valve Boundary Condition

The flow rate at the valve is

$$Q_P = -A_2 C_V + \sqrt{(A_2 C_V)^2 + 2 C_V A_1} \tag{14}$$

where $C_V = \frac{(Q_R \tau)^2}{2 H_R}$, $Q_R$, and $H_R$ are the flow rate and pressure of the valve when they flow steadily, and $\tau$ is the relative opening of the valve.

By using the above $C^+$ characteristic line equation, the pressure of the valve is obtained as follows:

$$H_{P1} = D_1 + D_2 Q_P \tag{15}$$

### 2.4.4. Pressure Flow Equation at the Pipe Bend

The pressure flow equation is similar to that used by the steady-state simulation except that fluid inertia is considered. The equation takes the form

$$\frac{r\theta}{A} \frac{dm_2}{dt} = P_1 - P_2 - \frac{K m_2 |m_2|}{2 \rho A^2} \tag{16}$$

where $r$ is the end radius on the centerline, $\theta$ is the bend angle, $A$ is the bend cross-sectional area, $\frac{dm_2}{dt}$ is the change of mass flow rate to node 2, $P_1$ is the pressure at the inlet to the bend, $P_2$ is the pressure at the outlet of the bend, $K$ is the corrected bend loss coefficient, and $\rho$ is the fluid density.

### 2.5. Flowmaster Calculation Model

Flowmaster software can build models based on user needs not only by adding the fluid software for analytical calculations, but also by modeling complex fluid systems. Flowmaster plays a crucial role

in a wide range of fields and is widely used for various fluid systems such as energy power, aerospace, automotive, marine, municipal, and other industries.

In this study, Flowmaster software is used to simulate the impacts of parameters such as the bending coefficient R/D, leakage hole size, and leakage position on the transient hydraulic characteristics of the pipe when a leak occurs. A simplified model of water pipeline with bending angles is shown in Figure 3. The total length of the pipeline is 100 m, with an effective pipe length of 90 m, a pipe diameter of 0.04 m, and a pipe wall (inner wall) absolute roughness of 0.025 mm. The downstream reservoir water level is 2 m, the upstream water head is 50 m, and a valve is installed at the position of 90 m of the pipeline. The velocity of the water hammer wave in the pipeline is 1000 m/s. The pressure outside the leak hole is atmospheric pressure. The calculation model is shown in Figure 4.

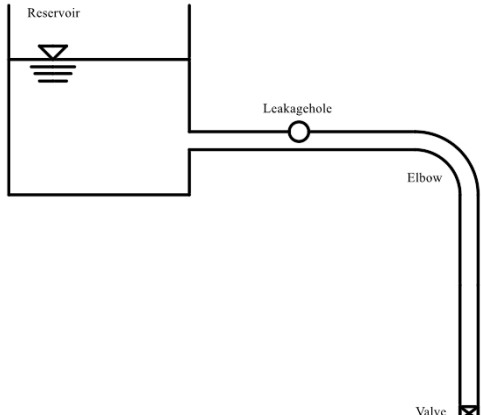

**Figure 3.** Simplified model of a water pipeline with bending angles.

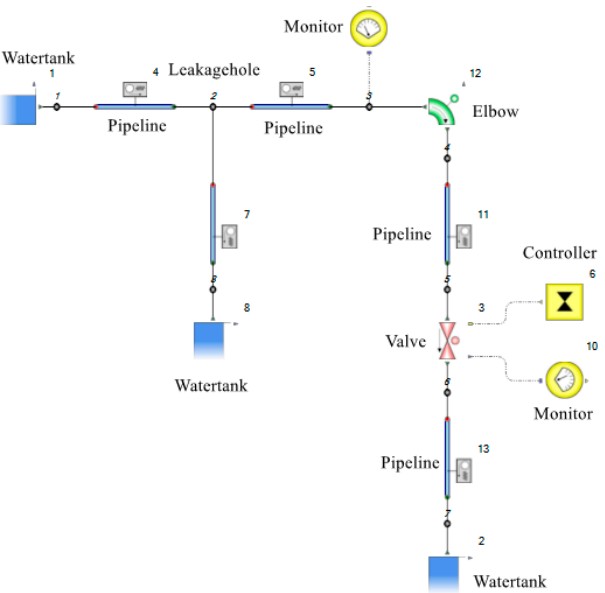

**Figure 4.** Simulation model of a pipe leakage with an elbow.

As shown in Figure 4, component 1 is the upstream reservoir, component 2 is the downstream reservoir, components 5, 4, 11, and 13 are elastic pipes, component 3 is a ball valve, and component 6 is the controller whose function is to control the switching of the valve. Component 12 is an elbow defined by the ratio of the bending radius *R* to the inner diameter *D* of the elbow, which is set at 70 m of the pipe, with a roughness of 0.25 mm, and an inner diameter of 0.04 m. The valve end monitoring node is 6, and the leakage node is 2. When a leak occurs, pipe element 7, which has a length of 1 mm and an inner diameter ad (ad = 0, 1, 2, 3 mm), is connected at 40 m at node 2 of the pipe to simulate

leakage, and 0.002 s is set to calculate the time step. The monitored data mainly includes the end of the valve at node 6 and the pressure change in leaking node 2, including the inflow of element 4 and the leakage flow rate of element 7.

## 3. Model Validation

In [31], a pipeline transient flow model was established considering unsteady friction, and this was compared with the experimental results of Guo [32]. The data were shown to be in good agreement. To verify the accuracy of the simulation, the simulation results of Flowmaster were compared with the experimental results. This comparison is shown in Figure 5.

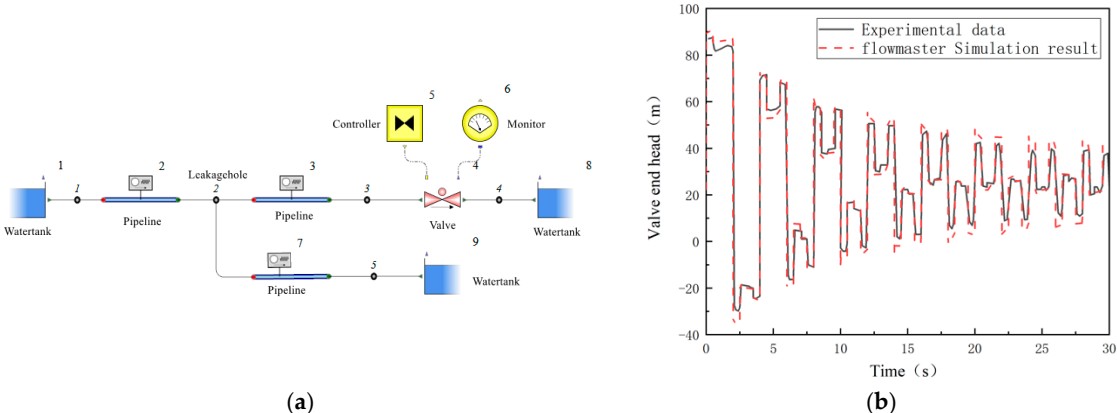

(**a**)　　　　　　　　　　　　　　　　　　　　　　　　　(**b**)

**Figure 5.** Comparison of the model experiment and simulation results of straight pipe leakage. (**a**) Leakage model of straight pipeline; (**b**) Pressure curve at the end of the pipeline.

It can be seen from Figure 5 that the results of pipeline leakage caused by the pressure size and drop position are consistent, and the pressure curve attenuation speed trend and cycle of the consistent degree are very high. Owing to the difference between the leakage hole and the short tube model formed with Flowmaster software and the complex physical characteristics of the components of the pipeline, the software simulation has greater ability than the experimental data to calculate the pressure of the valve; however, the maximum relative error is very small, only 3.2%.

## 4. Results and Discussion

### 4.1. Transient Characteristics of Leakage with a 90° Elbow Water Pipeline

Flowmaster software is used to simulate different bending coefficients R/D with and without leakage, and the influences of different leakage holes and leakage positions on the transient characteristics of the tube are compared and analyzed.

### 4.1.1. Effect of R/D without Leakage

In the absence of leakage, the influence of the bending coefficient on the flow in the pipe tube is analyzed. The values of bending coefficient R/D are 1, 2, and 3 respectively, and the curve of changes in valve end pressure and upstream flow in the pipeline is obtained, as shown in Figure 6.

By comparing Figures 6a and 6c, it can be observed that the overall cycle of change and the trend of changes in flow and end pressure do not vary with the bending coefficient R/D. However, it can be seen from the local variations shown in Figure 6b,d that the amplitude of the abrupt point is affected by the bending coefficient in the abrupt range of the flow rate and the end pressure amplitude, and this is proportional to the bending coefficient R/D.

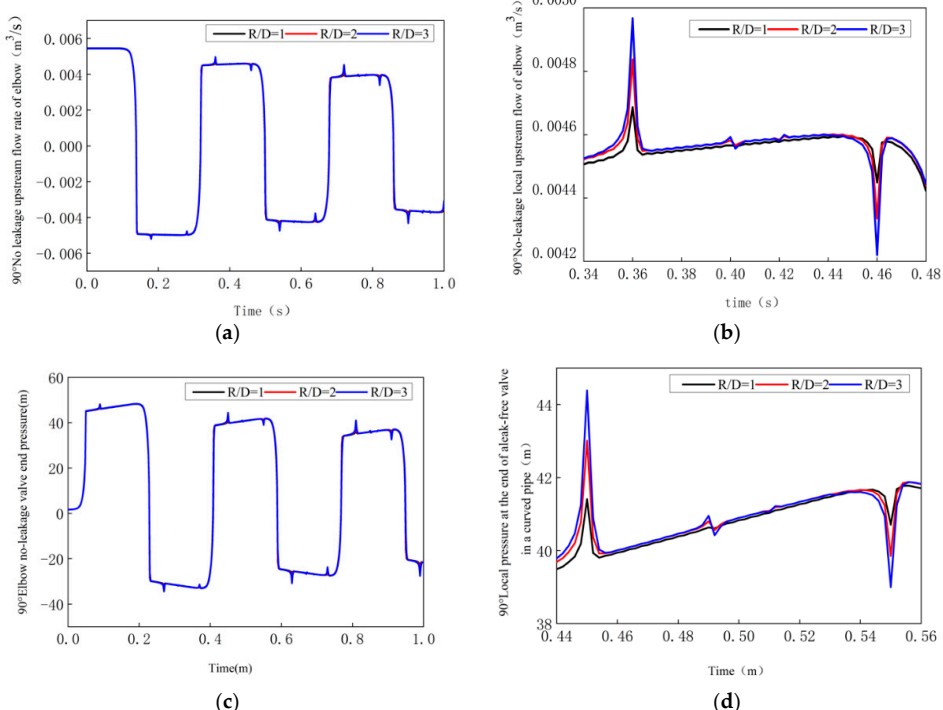

**Figure 6.** Effects of R/D (R is the turning radius of pipe, D is the inner diameter of pipe) on the transient characteristics of pipe with 90° elbow. (**a**) Upstream flow without leakage; (**b**) Local variations in upstream flow without leakage; (**c**) Changes in valve end pressure without leakage; (**d**) Local variations in valve end pressure without leakage.

### 4.1.2. Influences of R/D When a Leak Occurs

In this study, the leakage aperture ad is 1 mm, the leakage location is 40 m from the upstream reservoir, and the values of the bending coefficient R/D are 1, 2, and 3. The hydraulic variation characteristics in the 90° elbow pipe are simulated considering the occurrence of a leak. The trends of the valve end pressure, upstream flow, leakage flow, and leakage point pressure are mainly analyzed, as shown in Figure 7.

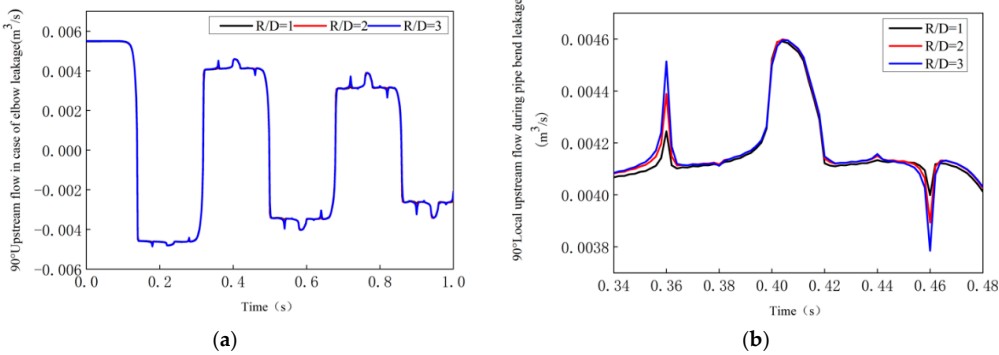

**Figure 7.** *Cont*.

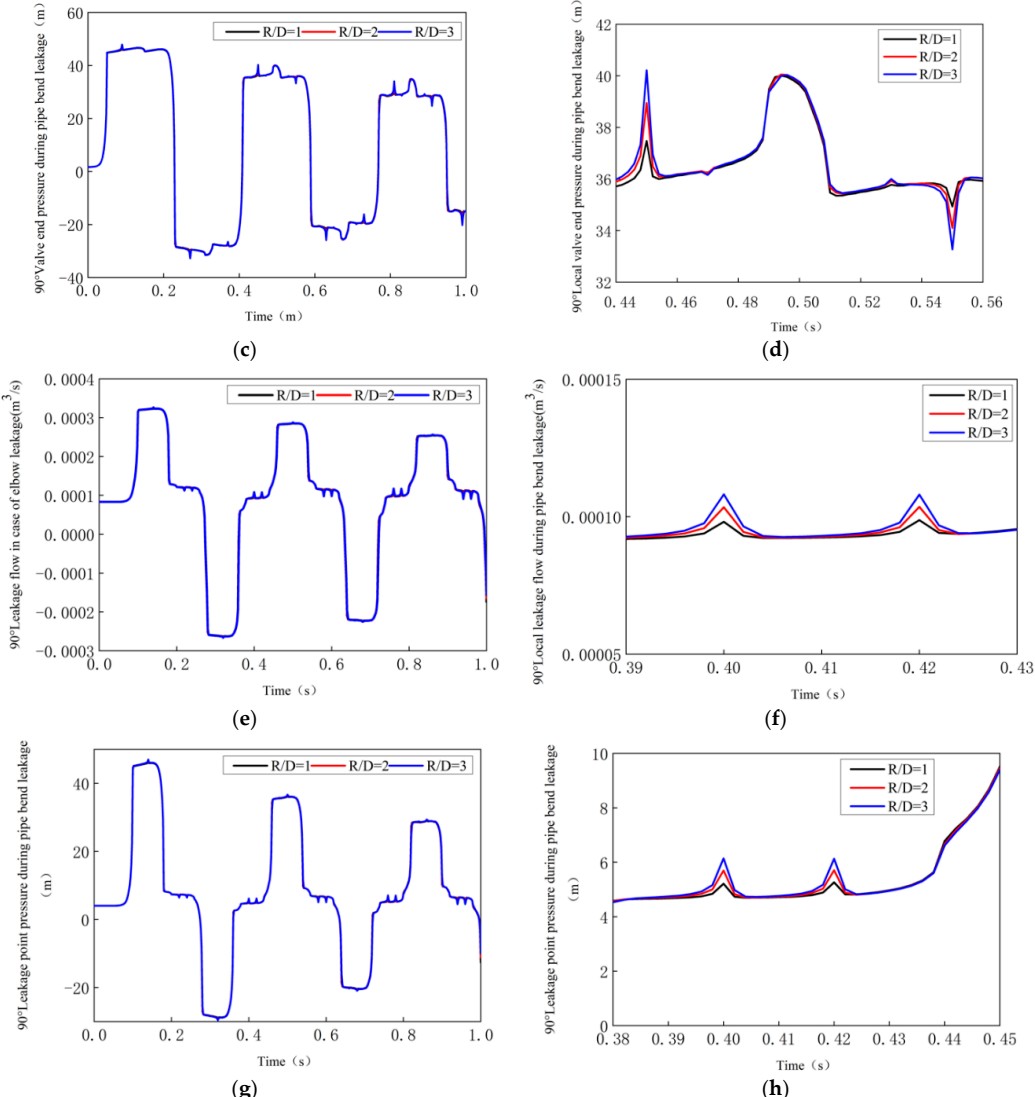

**Figure 7.** Effects of R/D on the parameters of a leaking pipe with a 90° elbow. (**a**) Changes in upstream flow when a leak occurs; (**b**) Local changes in upstream flow when a leak occurs; (**c**) Changes in valve end pressure during leakage; (**d**) Local changes in pressure at the end of the valve when a leak occurs; (**e**) Changes in leakage flow when a leak occurs; (**f**) Local changes in leakage flow when a leak occurs; (**g**) Changes in leakage point pressure when a leak occurs; (**h**) Local changes in leakage point pressure with leakage.

It can be seen from Figure 7a–d that when leakage occurs, the pressure change at the elbow valve and the periodic attenuation of the upstream flow of the elbow are not affected by the bending coefficient R/D. The bending coefficient mainly affects the amplitude of the sudden change in the position of the curve. The larger the bending coefficient R/D, the larger the amplitude of the mutation position, and the sharper the amplitude. However, the bump at the midpoint of the amplitude is mainly caused by the leakage of the pipeline and has an insignificant relationship with the bending coefficient R/D; thus, the influence of R/D is almost negligible. Simultaneously, we also focused on monitoring the leakage pressure and leakage flow, as shown in Figure 7e–h. It was found that the influence of the bending coefficient R/D on the leakage valve pressure and upstream flow change is consistent, and the impact mainly occurs at the sudden change point.

### 4.1.3. Influences of the Leak Hole Size

It is assumed that the leakage occurred at 40 m from the upstream reservoir, the leakage aperture size ad are 1, 2, and 3 mm, and the fixed bending coefficient R/D is 1. The effects of the simulated leak hole size on the flow, pressure, and leakage flow at each point in the pipeline are shown in Figure 8.

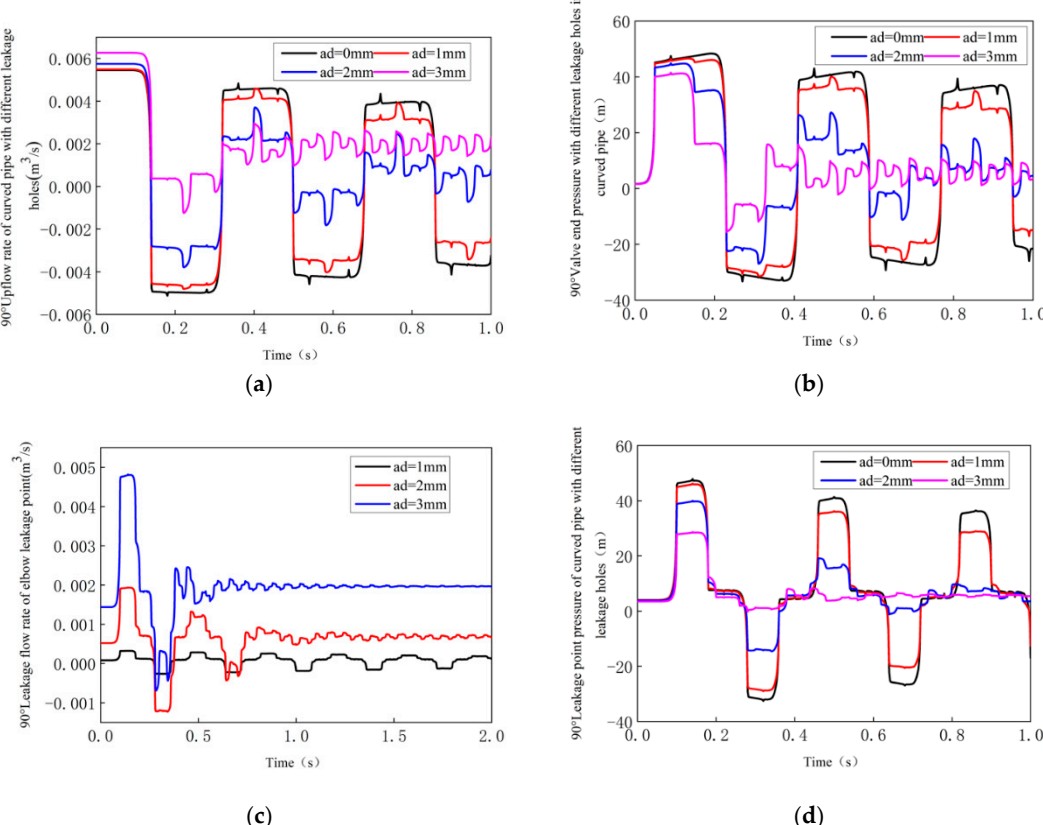

**Figure 8.** Effects of leak size on the parameters of a pipe with a 90° elbow. (**a**) Changes in upstream flow with different leak hole sizes; (**b**) Changes in valve end pressure with different leak hole sizes; (**c**) Changes in leakage flow with different leak hole sizes; (**d**) Variations in leakage point pressure with different leak hole sizes.

It can be seen from the variation curve of Figure 8 that the leakage hole size is proportional to the leakage flow rate, and as the size of the leakage hole decreases, the leakage flow tends toward zero. Furthermore, the larger the upstream flow value of the leakage position and the pressure values at the valve and node, the slower the curve decays. When the sudden drop in the first peak of the pressure at the valve gradually decreases, the curve becomes smoother and more regular.

### 4.1.4. Influences of the Leakage Position

When the leakage occurs, it is assumed that the fixed bending coefficient R/D is 1, the leakage aperture ad is 1 mm, and the leakage positions are at 40 m, 60 m, and 80 m positions along the pipe. Flowmaster software is used to simulate the pressure and flow changes in the tube under the working conditions, and the results are shown in Figure 9.

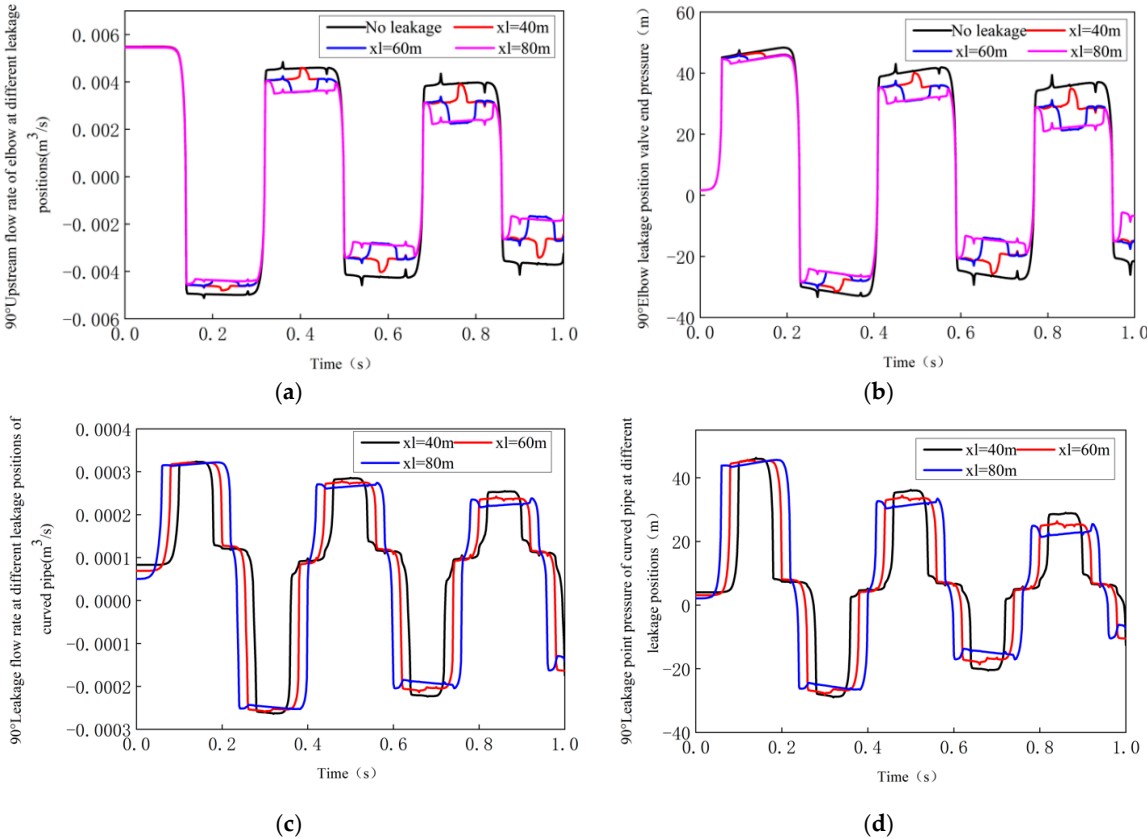

**Figure 9.** Effects of the leakage position on the parameters of a leaking pipe with a 90° elbow. (**a**) Changes in upstream flow at different leakage locations; (**b**) Changes in valve end pressure at different leakage locations; (**c**) Changes in leakage flow at different leakage locations; (**d**) Variations in leakage point pressure at different leakage locations.

It can be seen from Figure 9 that the farther the leakage position is from the valve, the larger the amplitude of the valve end pressure and upstream flow curve, and the symmetrical fluctuation phenomenon occurs. However, the leakage flow rate and leakage point pressure are not significantly changed.

## 4.2. Influence of Different Angles of the Bending Pipe on the Hydraulic Characteristics of a Pipeline

To better understand the impact of the bend angle on the leakage of the water pipeline, simulations of the straight pipe, 90° elbow, and 180° elbow in the presence or absence of leakage are carried out. In addition, the flow rate and pressure change of different bending pipe angles during leakage are analyzed.

### 4.2.1. Influence of a Bent Pipe in a Pipeline without Leakage

In the absence of leakage, it is assumed that the elbow is located at the position of 70 m, and the bending coefficient R/D of the elbow is 1. The curves of the end pressure and upstream flow of the straight pipe, 90° elbow, and 180° elbow are calculated, as shown in Figure 10.

It can be seen from Figure 10 that the presence of the elbow causes the water shock wave to refract and reflect at the elbow, resulting in an increase in fluid loss within the pipe, and the pipe pressure and flow rate with the elbow are generally less than the straight pipe parameters. The parameters in the tube are inversely proportional to the bending angle of the elbow. The presence of the elbow causes distortion of the amplitude; however, the magnitude of the distortion, and the trend of the curve and decay rate are not affected by the bending angle.

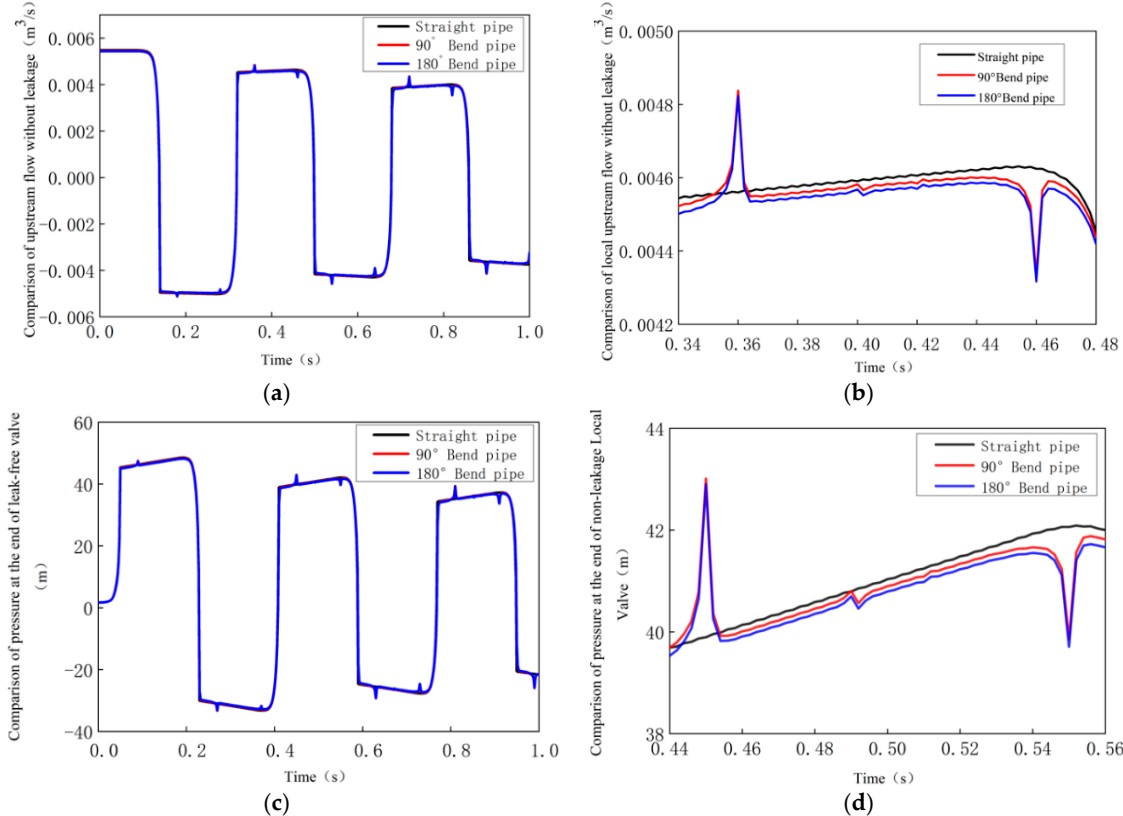

**Figure 10.** Influences of the bending angle on the flow and pressure in a pipe without leakage. (**a**) Effect of bending angle on upstream flow without leakage; (**b**) Effect of bending angle on local upstream flow without leakage; (**c**) Comparison of valve end pressure without leakage; (**d**) Effect of bending angle on the end pressure of local valve without leakage.

### 4.2.2. Influence of the Bending Angle on Pipeline during Leakage

It is assumed that the leakage aperture ad is 1 mm and the bending coefficient R/D is 1 at the time of the leakage. The leakage occurred at a distance of 40 m from the upstream reservoir, and the simulation determined the changes in the end pressure and upstream flow of the straight pipe, 90° elbow, and 180° elbow, as shown in Figure 11.

It can be seen from Figure 11 that the influences of the bending angle on the flow rate and pressure changes in the leaking pipe are similar, irrespective of whether or not there is a leak. However, as the elbow increases the coefficient of friction resistance of the pipe, the total energy carried by the fluid in the pipe decreases, such that the valve pressure and inlet flow rate with the elbow become smaller than those in pipes without the elbow. Owing to the influence of the leak hole on the water hammer wave, the changes in the pressure amplitude and flow rate during the leak are more severe and apparent than when there is no leakage. There is a mutation zone in the middle of the parameter amplitude, and the amplitude parameter of the abrupt zone is smaller due to the presence of the elbow. When the leakage occurs, the magnitude of the sudden change at the 90° bending angle is lower than at the 180° bending angle.

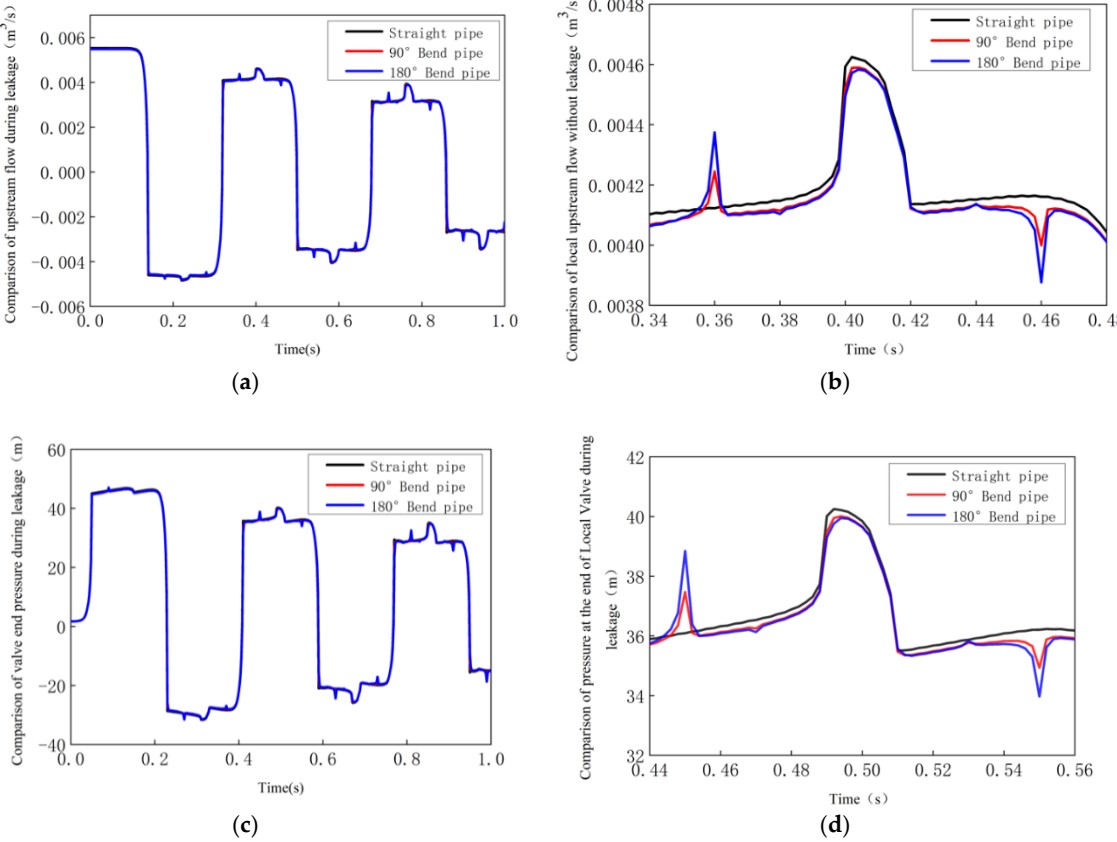

**Figure 11.** Influences of the bending angle on flow and pressure in a leaking pipe. (**a**) Effect of bending angle on upstream flow during leakage; (**b**) Local changes in upstream flow at the time of leakage; (**c**) Effect of bending angle on valve end pressure during leakage; (**d**) Local variations in valve end pressure during leakage

## 5. Conclusions

In this study, the transient hydraulic process of pipeline leakage is simulated accurately using Flowmaster software. The influences of parameters such as bending coefficient R/D, different leakage holes, and leakage position on the transient flow law of the pipeline with or without leakage are analyzed. The hydraulic characteristics of a straight pipe and bent pipes with 90° and 180° elbows are compared and analyzed in the absence of leakage. The concrete conclusions are as follows:

(1) Irrespective of whether a leak occurs or not, the overall period and variation trend of the flow rate and end pressure in the tube do not vary depending on the bending coefficient R/D. The bending coefficient mainly changes the magnitude of the sudden change point, and hence, the amplitude of the sudden change point is proportional to R/D.

(2) The size of the leak hole is proportional to the amount of leakage. As the leak hole gradually decreases, the leakage flow gradually approaches zero flow. The greater the upstream flow value at the leakage, the pressure at the valve, and the pressure at the node are, the slower the curve decays.

(3) The location of the leakage has a significant impact on the transient hydraulic characteristics of the tube. The farther the leak position is from the valve, the greater the amplitude of the valve end pressure and the upstream flow curve. A symmetrical wave phenomenon occurs, whereby the leak position of the pipe can be located.

(4) The influence of the elbows with different bending angles on the parameters of the leaking pipe is almost the same as when there is no leakage. Furthermore, the degree of amplitude change when the bending angle is 90° is significantly lower than the sudden amplitude change of the 180° elbow.

**Author Contributions:** Q.Z. collected original data; F.W. and Z.Y. developed the numerical model and performed the simulations; Q.Z., G.L. and J.Z. analyzed the results and drafted the manuscript; all authors contributed to the review and editing of the manuscript.

**Funding:** This research was funded by the Water Conservancy Science and Technology Project of Shaanxi Province (Grant No. 2016slkj-14), Independent Research Project of the State Key Laboratory of Ecological and Water Conservancy in the Northwest Arid Region (Grant No. 2017ZZKT-1), the National Natural Science Foundation of China (Grant No. 51906201, 51706180, and 11605136), and the Education Scientific Research Project for the Education Department of Shaanxi Province (Grant No. 16JK1542 and 16JK1088).

**Acknowledgments:** This research received no external funding.

**Conflicts of Interest:** The authors declare no conflicts of interest.

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
