# Peer review of "Simulation of the Transient Characteristics of Water Pipeline Leakage with Different Bending Angles"

_water, doi:10.3390/w11091871_

Round 1
Reviewer 1 Report
This paper is very interesting and fits the aims of the Journal. It is well written and the obtained results confirm that transient test-based techniques (TTBTs) are a promising tool for fault detection. However, before it can be published, the below requests should be addressed (my aim is to improve an already good paper!).
Literature review (in the Introduction)
It is very good and very interesting papers are cited. However, it can be improved by considering that:
within TTBTs – particularly in real pipe systems – the technique used to generate pressure wave plays a crucial role. Such an aspect should be mentioned in the revised version. In my opinion [ar1], and [ar2] – where some different techniques are used – could help; it is very important to point out differences between Transmission Mains (TMs) and Water Distributions Networks (WDNs). In fact, the presence of many branches in WDNs makes very difficult the use of TTBTs. In my opinion [ar3], and [ar4] – where this problem is discussed are used – could help; row 40: the use of the damping of pressure peaks to locate and size leaks is very interesting. Maybe [ar5] could enrich the discussion.
Numerical model
Since Authors consider very fast transients, it is very important to give more details about the model used in Flowmaster to simulate unsteady friction (by the way, at row 105, I would speak in terms of Js = steady-state component and Ju = unsteady-state component or unsteady friction (UF)). In my opinion [ar6] could help.
The executed analysis of the factors that influence the transient response of a leaky pipe is very well done. However, the role of the initial pressure at the leak should be highlighted. With regard to such an aspect, in my opinion, [ar7] could help.
Additional References
[ar1] Shucksmith, J.D., Boxall, J.B., Staszewski, W.J., Seth, A., and Beck, S.B.M. Onsite leak location in a pipe network by cepstrum analysis of pressure transients. J. AWWA 104(8), 2012, E457-E465.
[ar2] Lee P.J., Lambert M.F., Simpson A.R. and Vítkovský, J.P. (2006). “Experimental verification of the frequency response method of leak detection.” J. Hydraulic Research, 44 (5), 693-707.
[ar3] Laven, L. & Lambert, A.O. (2012). What do we Know About Real Losses on Transmission Mains? Proceeding, IWA Specialised Conference Water Loss, Manila (RP), 1-10.
[ar4] Meniconi, S., Brunone, B., and Frisinghelli, M. (2018). On the role of minor branches, energy dissipation, and small defects in the transient response of transmission mains. Water, 10(2), 187.
[ar5] Brunone, B., Meniconi, S., and Capponi, C. (2019). Numerical analysis of the transient pressure damping in a single polymeric pipe with a leak. Urban Water Journal, 15(8), 760–768.
[ar6] Duan, H. F., Ghidaoui, M. S., Lee, P. J., and Tung, Y. K. (2012). “Relevance of unsteady friction to pipe size and length in pipe fluid transients.” J. Hydraul. Eng., 10.1061/(ASCE)HY.1943-7900.0000497, 154–166.
[ar7] Liou C (1998) Pipeline leak detection by impulse response extraction. J Hydraul Eng. 120:833–838.
Author Response
Dear Editors and Reviewers,
Thank you for your letter and the reviewers’ comments concerning our manuscript. Those comments are all valuable and very helpful for revising and improving our paper, as well as the important guiding significance to our researches.We have studied the comments carefully and made relevant revisions marked in red in the paper. We hope the revised manuscript would meet with your approval. The main corrections in the paper and the detailed responses to the reviewer’s comments are the attachment. Please see the attachment.
We tried our best to improve the manuscript and made some changes in the manuscript. These changes will not influence the content and framework of the paper.
We appreciate for Editors/Reviewers’ hard work and valuable suggestions earnestly, and hope that the corrections will be satisfying.
Once again, thank you very much for your comments and suggestions.

Reviewer 2 Report
The paper refers to an analysis of the effect of leakage on transient flow in pipe with an elbow with different bending angles and bending ratio. The authors use the Flowmaster code to analyse the effect of different parameters, without any comparison with experimental results. Although the significance of the model is limited by the absence of the latter aspect, the paper can presents an interest. But the authors have to consider the following points before publication.
The sentence “… many domestic and foreign scholars …” (line 35) should be rewritten “… many scholars …”. The sentence “… the continuous equation …” (line 103) should be rewritten “… the continuity equation …”. Lines 103-106: please omit the word “variable” (time, velocity, displacement). Line 105: “displacement” should be redefined as “distance along the pipe”. Lines 105-106: “constant friction” and “non-constant friction” should be redefined respectively as “steady friction” and “unsteady friction”. In Eq. (2) the terms of steady friction and unsteady friction are indicated, but their expressions are not defined. The authors present the results of the analysis on transient of the characteristics of elbow and leakage without any presentation of the equations used to model these aspects. This is a fundamental point: all the equations of the model have to be presented and discussed. Due to the absence of comparison with experimental results, the model is not properly validated, and then the authors should specify that the numerical results have to be considered with caution.Author Response
Dear Editors and Reviewers,
Thank you for your letter and the reviewers’ comments concerning our manuscript. Those comments are all valuable and very helpful for revising and improving our paper, as well as the important guiding significance to our researches.We have studied the comments carefully and made relevant revisions marked in red in the paper. We hope the revised manuscript would meet with your approval. The main corrections in the paper and the detailed responses to the reviewer’s comments are the attachment. Please see the attachment.
We tried our best to improve the manuscript and made some changes in the manuscript. These changes will not influence the content and framework of the paper.
We appreciate for Editors/Reviewers’ hard work and valuable suggestions earnestly, and hope that the corrections will be satisfying.
Once again, thank you very much for your comments and suggestions.
